# A UAV Thermal Imaging Format Conversion System and Its Application in Mosaic Surface Microthermal Environment Analysis

**DOI:** 10.3390/s24196267

**Published:** 2024-09-27

**Authors:** Lu Jiang, Haitao Zhao, Biao Cao, Wei He, Zengxin Yun, Chen Cheng

**Affiliations:** 1School of Internet of Things, Nanjing University of Posts and Telecommunications, Nanjing 210003, China; jianglu@smail.nju.edu.cn; 2State Key Laboratory of Remote Sensing Science, Aerospace Information Research Institute, Chinese Academy of Sciences, Beijing 100101, China; 3State Key Laboratory of Remote Sensing Science, Innovation Research Center of Satellite Application, Faculty of Geographical Science, Beijing Normal University, Beijing 100875, China; caobiao@bnu.edu.cn; 4Zhejiang Carbon Neutral Innovation Institute, Zhejiang University of Technology, Hangzhou 310014, China; wei.he@zjut.edu.cn; 5School of Software Engineering, Jinling Institute of Technology, Nanjing 211169, China; yunzengxin@jit.edu.cn; 6Hangzhou Zhiyu Space Planning and Design Co., Ltd., Hangzhou 310030, China; cerochan@163.com

**Keywords:** thermal infrared remote sensing, land surface temperature, unmanned aerial vehicle, format conversion, surface microthermal environment

## Abstract

UAV thermal infrared remote sensing technology, with its high flexibility and high temporal and spatial resolution, is crucial for understanding surface microthermal environments. Despite DJI Drones’ industry-leading position, the JPG format of their thermal images limits direct image stitching and further analysis, hindering their broad application. To address this, a format conversion system, ThermoSwitcher, was developed for DJI thermal JPG images, and this system was applied to surface microthermal environment analysis, taking two regions with various local zones in Nanjing as the research area. The results showed that ThermoSwitcher can quickly and losslessly convert thermal JPG images to the Geotiff format, which is further convenient for producing image mosaics and for local temperature extraction. The results also indicated significant heterogeneity in the study area’s temperature distribution, with high temperatures concentrated on sunlit artificial surfaces, and low temperatures corresponding to building shadows, dense vegetation, and water areas. The temperature distribution and change rates in different local zones were significantly influenced by surface cover type, material thermal properties, vegetation coverage, and building layout. Higher temperature change rates were observed in high-rise building and subway station areas, while lower rates were noted in water and vegetation-covered areas. Additionally, comparing the temperature distribution before and after image stitching revealed that the stitching process affected the temperature uniformity to some extent. The described format conversion system significantly enhances preprocessing efficiency, promoting advancements in drone remote sensing and refined surface microthermal environment research.

## 1. Introduction

The surface thermal environment (STE) results from interactions among the surface, atmosphere, and human activities, impacting climate change, energy use, and human health [1,2,3,4,5]. Urbanization has replaced natural surfaces with impermeable, complex, and dense artificial surfaces, disrupting heat balance and increasing sensible heat while decreasing latent heat, leading to significant changes in the STE, notably, to the urban heat island (UHI) effect [6,7,8]. As demands for comfort, health, and safety rise, understanding and improving the STE becomes increasingly urgent [8,9,10,11].

Thermal infrared remote sensing is an effective method for directly observing surface temperatures on regional and global scales [12] and has been widely used to study the spatial patterns and temporal evolution of surface thermal environments, especially the UHI effect [13,14,15]. Traditional research often calculates UHI intensity by subtracting the suburban surface temperatures from the urban ones, exploring urban–suburban thermal differences [16,17]. However, this method is overly simplistic, neglecting differences in three-dimensional structures and surface materials within urban and suburban areas, thus affecting the accurate STE assessment [18]. Increasing attention is now given to the micro-scale STE, with research shifting from urban regions to local areas [4,19]. The local climate zone (LCZ) classification system emerged to address this [20]. Based on surface cover, structure, materials, and human activities, the LCZ system established 10 quantitative indicators to divide the surface into 17 types [20]. The LCZ scheme provides detailed descriptions of the thermal environment of homogeneous surfaces and is widely used in micro-scale studies [21,22,23]. These studies often rely on satellite remote sensing data and face limitations in time and spatial resolution, which makes it difficult to capture fine-scale changes in heterogeneous STEs. For example, the spatial resolutions of commonly used Moderate Resolution Imaging Spectroradiometer (MODIS) and Landsat Thematic Mapper (TM) surface temperature products are 1 km and 120 m, respectively [17,24,25,26].

With the advancement of unmanned aerial vehicle (UAV) technology, characterized by low cost, high flexibility, high resolution, and high revisit frequency, the limitations of current satellite thermal infrared remote sensing are poised to be addressed. Recently, UAV remote sensing has become crucial for detailed STE studies [19,27,28,29,30,31]. DJI drones lead the UAV industry, being equipped with professional flight control programs and high-precision RTK modules for sampling surface temperatures and recording flight and geographic data. They feature advanced thermal infrared sensors, such as the Zenmuse XT and XT2 (DJI, Shenzhen, China, powered by FLIR) and the DJI self-developed Zenmuse H20T, H20N, Matrice 30T, Mavic 2EA, and 3T sensors (DJI, Shenzhen, China), providing high-quality thermal images (centimeter-level spatial resolution, half-hour temporal resolution, ±2 °C accuracy). DJI self-developed sensors are highly integrated into drone systems, ensuring stability and supporting extensive mission planning and data analysis. However, they have limitations in image stitching and processing. FLIR sensors output TIFF and radiometric JPEG (RJPEG) formats, compatible with most processing software (e.g., ContextCapture (v10.20.0), Pix4dMapper (v4.4.12), PhotoScan (v1.2.7)) [19,32,33]. Although DJI self-developed sensors also use radiometric JPEG (file suffix *.JPG), carrying temperature and geographic data, they are essentially RGB format files and lose temperature information in the existing processing software. Single JPG images can be analyzed with the DJI Thermal Analysis Tool (v3.0) (https://www.dji.com/). For stitching multiple JPG images, technical challenges exist. Although the DJI thermal software development kit (TSDK) (v1.5) (https://www.dji.com/) can convert JPG images to the RAW format, retaining temperature data, geographic information is lost, hindering extensive analysis in image stitching and processing software. This limits the broad application of UAV thermal infrared remote sensing in STE research.

In recent years, the use of DJI drones with a thermal sensor for STE research has increased. To achieve thermal image stitching, two main methods are commonly used. One involves format conversion software [27,34,35,36], such as ATygeo Thermal (v2.0) (www.atyges.es/en/product/atygeo-thermal/) and ThermoConverter (v1.8.0) (www.thermoconverter.com), which convert JPG images to the TIFF format (containing temperature and geographic information) for easier stitching. However, these commercial software options are expensive, which limits their use. The other method separates temperature and geographic information, inputting it separately into the stitching software [29,37,38]. JPG thermal images are converted to the TIFF format (losing sensor parameters and geographic information), and other software (e.g., DJI Terra) extracts geographic information from the JPG images for input into a stitching software (e.g., Pix4dMapper (v4.4.12)). This method has drawbacks, including the loss of sensor parameters and flight angle information, which can affect the stitching accuracy, and complex processing steps, which increases error likelihood and reduces efficiency. A key issue in current research is the lack of simple, low-cost software or systems to losslessly convert JPG thermal images to the Geotiff format. Additionally, high-spatial-resolution UAV thermal images are needed for detailed spatiotemporal studies of the local/micro-surface thermal environment.

We addressed these issues in this study. The main contribution of this paper includes two aspects: (1) this paper developed a format conversion system, ThermoSwitcher, for processing DJI drone thermal JPG images, enabling lossless conversion to the Geotiff format; (2) ThermoSwitcher was applied to analyze the urban thermal environment and explore the spatiotemporal variations in surface temperature from a high-resolution perspective. Specifically, two study areas in Nanjing with various local zones were selected. We used a DJI drone with a thermal sensor to collect surface thermal images; then, ThermoSwitcher performed format conversion, Pix4dMapper (v4.4.12) handled image stitching, and ArcMap (v10.8) facilitated sub-area surface temperature extraction and statistics. This enabled the analysis of the spatial and temporal characteristics and differences in thermal conditions across different local zones. Overall, ThermoSwitcher provides convenience and support for preprocessing DJI thermal images, potentially advancing fine-scale studies on surface microthermal environments using UAV remote sensing.

This paper is organized as follows: Section 2 describes the methods and data of this paper, including the design and operation mechanism of ThermoSwitcher, a UAV thermal infrared image format conversion system, the study area selected to carry out the spatiotemporal analysis of local thermal environments, as well as the process of experimental collection and data processing; Section 3 shows the application of ThermoSwitcher in the analysis of local microthermal environments, where we analyzed the collected data and compared the differences in the spatiotemporal distribution of different local area thermal environments, as well as the temperatures before and after the splicing of the UAV thermal infrared images; Section 4 discusses the strengths and weaknesses of this paper, as well as the outlook for future research work; and finally, Section 5 provides a short summary and some perspectives.

## 2. Methods and Data

### 2.1. Format Conversion System

To achieve the lossless conversion of JPG format thermal images to the Geotiff format, we designed a system called ThermoSwitcher (Figure 1). It should be noted that the DJI-developed thermal sensors have preset environmental parameters that may affect temperature measurements. Therefore, calibration is needed during data preprocessing based on the actual sampling scenario (https://www.dji.com/). The four calibration parameters are the following: (1) Distance: Range, 1 to 25 m. Enter 25 m if it exceeds this range. The default calibration distance is usually 5 m, where measurements are most accurate. Distances too close or far increase errors. (2) Humidity: Range, 20 to 100%. The default value is usually 50% or 70%, depending on the sensor model, and should be set according to the actual environment. Accurate settings have a minor impact on precision. (3) Emissivity: Range, 0.10 to 1.00. The default value is usually 0.95 or 1.00, depending on the sensor model. Set it based on common material emissivity tables and the surface cover type. Accurate settings significantly affect precision. (4) Reflection temperature: Range, −40.0 to 500.0 °C. The surrounding objects’ energy may be reflected by the target surface and received by the sensor, causing errors. The default value is usually 23.0 °C or 25.0 °C, depending on the sensor model. Generally, set it to ambient air temperature if there are no special high- or low-temperature objects. Accurate settings affect precision, with greater deviations causing larger impacts.

The workflow of ThermoSwitcher includes three steps (Figure 1a):

Step #1: Convert the original thermal JPG image to a RAW format image with temperature information. Use the ‘dji_irp.exe’ program from DJI Thermal SDK (https://www.dji.com/) to convert the JPG image to a RAW format image (32-bit float), based on the calibration parameters sampled during the measurement experiment. Compared with the thermal JPG format file, the RAW format file is still an image, and the resolution is the same as that of the original JPG image. The only difference is that the data for each pixel in this image no longer represent color information but rather correspond to temperature. Note that this step involves a call to the DJI Thermal SDK (call code example: “…/dji_irp.exe -s …/DJI_0001_R.JPG -a measure -o …/DJI_0001_R.raw --distance 25.0 --emissivity 0.96 --humidity 40 --reflection 30.0 --measurefmt float32”, note that the ellipsis represents the file path). Due to the commercial confidentiality agreement, the source code of the DJI Thermal SDK is not disclosed, and the specific mathematical formulas of the four parameters involved in temperature correction cannot be shown.

Step #2: Convert the RAW format image to a TIFF format image with temperature information. Read the temperature data from the RAW image and write it into a new TIFF image.

Step #3: Add geographic information to the TIFF image to create a Geotiff image. Read the EXIF information from the JPG image to obtain the sensor parameters and GPS coordinates, then write these data into the TIFF image to create a Geotiff image.

The ThermoSwitcher interface is depicted in Figure 1b. Set the file paths for the original JPG and the converted Geotiff images, configure the temperature calibration parameters based on the observed environmental variables, and perform the conversion. Note that this system and the user’s manual are provided as Appendix A, free of charge, for researchers to facilitate the processing of UAV thermal infrared imagery and to expand its potential research and applications.

### 2.2. Study Area

To verify the application ability of ThermoSwitcher in surface microthermal environment analysis, two regions in the suburbs of Nanjing, China, (32.13° E, 118.97° N) were selected as the area of interest (AOI), i.e., AOI_A and AOI_B (Figure 2). The region has a subtropical monsoon climate with distinct seasons. AOI_A and AOI_B cover 220 × 2000 m and 410 × 1800 m, respectively, and are oriented to the northwest–southeast (Figure 2a).

The land use types of the study areas are diverse. AOI_A and AOI_B are divided into six local interest zones each (Figure 2b). Specifically, zone A1 is a high-rise residential area with 25-to-27-story buildings (mainly concrete) and 46% vegetation cover, primarily evergreen trees, shrubs, and grass. Zone A2 is a mid-rise residential area with 11-story buildings and 45% vegetation cover, mainly evergreen trees and shrubs. Zone A3 is a river area with bare soil and 15% vegetation cover. Zone A4 is a metro station area with buildings and concrete/asphalt surfaces and 5% vegetation cover. Zone A5 is a mountainous forest area with 99% vegetation cover, mainly trees. Zone A6 is a low-rise residential area with 3-story buildings and 45% vegetation cover, mainly evergreen trees and shrubs. Zone B1 is a high-rise residential area with 17-to-18-story buildings and 45% vegetation cover. Zone B2 is a school area with 5-story buildings and large playgrounds, with 15% vegetation cover, mainly evergreen trees and shrubs. Zone B3 is a lake park with vegetation and water bodies and some bare soil. Zone B4 is a high-rise residential area with 20-to-34-story buildings and 45% vegetation cover. Zone B5 is a commercial area, similar to Zone A4, with buildings and concrete surfaces and 5% vegetation cover. Zone B6 is a river area, similar to Zone A3. Note that the vegetation cover values might vary due to different image collection times and were extracted using ImageJ software (v1.45) (https://imagej.nih.gov/ij/), a JAVA-based image processing tool with a simple visual interface.

### 2.3. Measurement Experiments and Data Processing

This study used a DJI M300 RTK quadcopter with Zenmuse H20T (DJI, Shenzhen, China) (Figure 3) to obtain thermal images. The Zenmuse H20T includes a thermal infrared lens, two visible (wide and zoom) lenses, and a laser rangefinder. The thermal infrared sensor has a resolution of 640 × 512 pixels, a pixel pitch of 12 μm, a wavelength range from 8 to 14 μm, a temperature measurement range from −40 °C to 150 °C, an accuracy of ±2 °C, and a sensitivity of <0.05 °C. For AOI_A and AOI_B, the UAV flight paths are shown in Figure 3d,e. The drone maintained a flight altitude of 300 m, thermal lens sampled at nadir direction with a spatial resolution of 0.26 m per pixel, and the flight speed was 13 m/s, with both 75% lateral and longitudinal overlap. The experiment was conducted on August 19, 2022, with two sampling sessions in the AOI_A area, each lasting about 8 min. The midpoint times were 15:30 (local time, 216 images collected) and 16:48 (197 images collected) (Table 1). Similarly, two sampling sessions were conducted in the AOI_B area, each lasting about 11 min. The midpoint times were 16:35 (284 images collected) and 18:35 (217 images collected) (Table 1). To ensure a quasi-synchronous surface temperature collection and reduce the impact of temporal changes, the number of images collected varied slightly depending on the drone battery life. The solar angles during the measurement are shown in Table 1; the weather was clear with light winds, and the maximum and minimum temperatures were about 35.7 °C and 31.2 °C, respectively. Considering the actual situation, the distance in ThermoSwitcher was set to the maximum (25 m), with an emissivity of 0.96 [39], and the air humidity and reflection/air temperature values were measured at a weather station (Figure 3c, refer to Ref. [40] for more details) about 1.5 km from the study area (Table 1).

The data processing workflow is as follows:

Step #1: Convert the DJI UAV thermal images from JPG to Geotiff format using ThermoSwitcher, retaining temperature and GPS information. The conversion took about 3 min on a standard laptop. Note that this study did not perform temporal normalization due to the short sampling duration, and the temperatures were surface brightness temperatures without emissivity separation and atmospheric correction.

Step #2: Mosaic the Geotiff thermal images using Pix4Dmapper (v4.4.12) to create a thermal image of the entire AOI_A and AOI_B, respectively.

Step #3: Clip the local zones and extract the surface temperatures using ArcGIS (v10.8). Import the stitched ortho-mosaic thermal images, clip them based on local zone boundary kml files, and extract the pixel temperature values for each zone for further analysis.

## 3. Results

### 3.1. Spatiotemporal Differences in Local Surface Brightness Temperatures

Based on the stitched ortho-mosaic thermal images, this section analyzes the spatial distribution and temporal variation patterns in different local zones (Figure 4 and Figure 5). The results revealed significant spatial heterogeneity across the whole study areas and notable spatiotemporal differences in surface temperatures across different local zones.

Regarding the overall spatial temperature distribution, high-temperature areas were mainly concentrated on sunlit artificial surfaces (e.g., asphalt roads and playgrounds), while low-temperature areas were found in shaded building areas, dense vegetation, and water bodies. This was primarily due to differences in the received solar radiation and the thermal properties of the surface materials. For instance, in the AOI_A study area at 15:30, the surface temperature difference exceeded 25.0 °C (Figure 4a), with the temperatures of asphalt roads, building rooftops, and subway station areas exceeding 45.0 °C, while vegetated and water body areas had temperatures below 20.0 °C. In more detail, taking advantage of the high-spatial-resolution images of UAVs, Figure 6 clearly shows the difference in the ground temperature of different features, including 10 vegetated features and 10 artificial features. The temperature distribution for the 10 vegetated features ranged from 28.1 °C (Blue No. 2, maple tree) to 35.7 °C (Blue No. 4, wilted grassland) (Figure 6), with an average temperature of 29.8 °C. Except for the withered flower (Blue No. 2) and withered grassland (Blue No. 4)—both of which were close to the bare soil—the temperatures of other vegetated features were concentrated between 28.1 °C and 29.6 °C. The temperature difference between the latter was relatively small (less than 1.5 °C). These subtle differences may be mainly related to vegetation type, leaf size, and spatial location. In contrast, the artificial features were hotter, and the temperature difference between them was greater. Specifically, the temperatures of the 10 artificial features were significantly higher than those of the vegetation features, ranging from 32.6 °C (Red No. 2, black metal car) to 44.1 °C (Red No. 6, glass roof) (Figure 6), with an average temperature of 36.8 °C. The metal vehicles (Red Nos. 2, 3, and 4) and the solar water heater (Red No. 9, mainly composed of a metal collector and glass) also had relatively similar temperatures (from 32.6 °C to 33.7 °C). The asphalt road, stone pavement, and concrete roof (Red Nos. 1, 5 and 7) had similar temperatures (from 39.2 °C to 40.7 °C). In contrast, the temperatures of the red-tile roof (Red No. 8) and red-brick pavement (Red No. 10) were slightly lower (37.0 °C and 34.2 °C, respectively), which may be related to their tilting angle, location and surroundings that cause them not always be in the optimal sun irradiation position.

There are significant differences in the spatial distribution and temporal variation rates of temperatures in different zones (Figure 4 and Figure 5). With regard to the spatial temperature differences, the zone mean temperature order was ranked, as exemplified by the 15:30 ZOI_A scenario, as follows: A4 (35.3 °C) > A2 (31.1 °C) > A1 (30.9 °C) > A6 (30.0 °C) > A3 (28.7 °C) > A5 (28.0 °C). These differences are related to surface coverage and material thermal properties. The subway station (zone A4) included materials like concrete and steel with low specific heat capacity, high thermal conductivity, and low infrared radiation efficiency, resulting in higher temperatures. Water bodies (zone A3) and mountain forests (zone A5) had lower temperatures. The differences in residential areas (zones A2, A1 and A6) may be due to variations in vegetation cover and building layout and height. The building layout affects heat circulation, and their height affects shading [4]. Moreover, except for the subway station (zone A4) and water bodies (zone A3, B6), the temperature in other zones was nearly normally distributed. This was due to the small size and sample number of the subway station and water bodies zones and the homogeneity of the material properties in the subway station zone (Figure 4). The water body areas included water, sparse vegetation, and bare soil, which led to distinct temperature frequencies with multiple peaks (Figure 4 and Figure 5). Larger zones with high heterogeneity, large thermal capacity, and dispersed heat sources showed nearly normally distributed temperatures. This phenomenon has important implications for remote sensing image analysis, reminding researchers to consider plot scale effects and environmental characteristics.

For the temperature variation rates, for AOI_A and AOI_B, the sampling intervals were of about 1 h, and the temperature differences (Δ*T*) between two samplings measured the temperature variation rates in different local zones. The Δ*T* ranking for AOI_A was A1 (2.3 °C) > A4 (2.0 °C) > A2 (1.4 °C) > A3 (1.1 °C) > A5 (0.5 °C) > A6 (0.4 °C). For AOI_B, it was B5 (1.8 °C) > B6 (1.6 °C) > B2 (1.5 °C) > B4 (1.4 °C) > B1 (1.1 °C) > B3 (0.8 °C). Although the observation times for AOI_A and AOI_B were different, making direct comparisons infeasible, the temperature variation rate differences within each zone shared similar causes. The variation rates were influenced by surface material properties, vegetation cover, building density and height, water bodies, topography, and microclimate interactions. Notably, due to a higher proportion of bare soil in the river zone B6 compared to A3, its temperature variation rate was significantly higher (Figure 5).

### 3.2. Temperature Comparison before and after Image Stitching

Taking zones A2 and B3 as examples, Figure 7 illustrates UAV thermal images, including raw and stitched data, with brightness temperature distribution histograms. Differences existed in the surface temperature before and after stitching, with varying temperature differences among zones. For zone A2, the original image shows residential building temperatures fluctuating between 20 °C and 45 °C (Figure 7a). Rooftops and artificial roads are high-temperature areas, while vegetated areas are cooler. After stitching, Figure 7c shows a more uniform temperature distribution, with rooftops still prominent but higher temperatures in roads and vegetated areas. Pixel temperature statistics (Figure 7e) indicates a wider range in the original image, with more points between 25 °C and 35 °C, averaging 27.9 °C. The stitched image reveals more concentrated pixel temperatures, with an average temperature of 30.6 °C (Figure 7e), showing smoother data and more prominent high-temperature areas. For zone B3, the original image shows concentrated lake temperatures between 20 °C and 25 °C, averaging 21.1 °C (Figure 7b,f). After stitching, the distribution remains concentrated, averaging 22.2 °C, but the artificial surface temperature in the lower right increases significantly (Figure 7d,f). Compared to zone A2, the lake temperature changes were smaller, indicating strong regulation, and smoothing had a little impact.

These results show that, although there were some temperature differences before and after stitching, significant temperature differences among different zones remained, not affecting the analysis of the temperature differences within the area (e.g., local urban heat islands). Smoothing and detail loss during stitching led to a more uniform temperature distribution, especially in building areas. Different area characteristics affected stitching differently. The building area temperature differences were more pronounced, indicating a significant increase in temperature after stitching, whereas the lake area, due to high specific heat capacity and temperature regulation, was less affected, maintaining a relatively small temperature distribution change.

## 4. Discussion

This paper focused on the development of a thermal image format conversion system for DJI drones, called ThermoSwitcher, which can losslessly convert thermal images in JPG format to the Geotiff format. This system addresses challenges in the stitching and processing of existing JPG thermal images. Additionally, the application of ThermoSwitcher was highlighted by analyzing the spatiotemporal characteristics of local surface thermal environments using UAV-based thermal images with sub-meter spatial resolution. The results demonstrated that UAV-based surface temperature data with sub-meter spatial resolution can capture fine surface details, allowing for the precise identification of subtle temperature gradients and localized urban heat island phenomena [30]. For instance, a UAV-based thermal sensor can capture temperature variations in narrow alleyways between buildings, rooftop green spaces, and areas near small water bodies, providing accurate data support for urban planning and environmental regulation. This paper primarily provides technical support for data processing (DJI thermal JPG image format converted to Geotiff format), and future research should pay more attention to the surface temperature correction accuracy of UAV thermal infrared images, improving data reliability [41,42]. In particular, future research should be carried out on how to improve the precision and accuracy of temperature measurements by corrected parameters, as well as on the design of an optimal observation scheme for UAVs (including weather conditions, equipment performance, flight altitude, flight speed, sampling frequency, flight path overlap, and so on) that may affect the image stitching quality. Moreover, future research can also focus on ThermoSwitcher’s application-related long-term observations and on the quantitative analysis of urban microthermal environments supported by UAV remote sensing. By establishing a database for regular drone monitoring, researchers can systematically track the dynamic changes of urban microthermal environments and identify seasonal and long-term trends. This has important implications for urban environmental management in the context of climate change. For example, research can help identify areas with severe heat island effects, providing data support for urban green space planning, building layout optimization, and energy efficiency improvement.

Furthermore, in this study, drones were used to collect surface temperature data in the research area four times, with each session lasting about 8 min. Due to the short collection duration, the spatiotemporal representativeness of surface temperature was not accounted for during image stitching. The issue of representativeness arises when the temporal or spatial resolution of an observation system is too high, leading to inconsistencies that may prevent the data from accurately representing the average conditions of the entire observed area or time period [43]. Generally, when sampling a large area, if the sampling duration exceeds 30 min, the representativeness issue should be carefully considered [39]. Neglecting this issue can result in systematic biases in surface temperature data, affecting the accuracy of the results. This problem is particularly significant in high-resolution thermal infrared remote sensing because the surface temperature often exhibits substantial heterogeneity across space and time. Specifically, future research should consider the following dimensions: (1) Temporal dimension: During drone-based surface temperature collection, significant changes may occur over time, influenced by dynamic factors such as solar radiation and surface heat exchange. Even within a relatively short period (e.g., 20 min), the temperature captured by images at different times may vary, especially under strong sunlight with rapid temperature fluctuations. As a result, the stitched images may present a biased temperature distribution, failing to accurately reflect the actual conditions of the study area. This issue has been widely reported in thermal infrared remote sensing. For instance, Lagouarde et al. [39] found that short-term temperature variations (from a few minutes to tens of minutes) can lead to significant differences, particularly in high-resolution remote sensing data. (2) Spatial dimension: Spatial heterogeneity in surface temperature means that small differences in surface features (such as vegetation, buildings, or bare ground) can lead to significant temperature variations in high-resolution thermal infrared images. With the high spatial resolution of drone remote sensing, the temperature measured by each pixel may only represent a very small area. If these small areas have significant spatial temperature differences that are not adequately addressed during image stitching, the resulting temperature image may not represent the average temperature distribution of the entire study area. Research by Hook et al. showed that surface temperature heterogeneity significantly affects temperature estimation accuracy in high-resolution contexts [44]. To mitigate these representativeness issues, we recommend further research into methods for effectively integrating multi-temporal and multi-spatial resolution data. This would reduce the impact of these issues. First, the time window for data collection during a single flight should be minimized, or multiple collections at the same location should be performed to reduce the impact of temporal temperature variations on the results. Time series analysis techniques, such as the DTC model [45], can be employed to weight the data collected at different times, providing more stable temperature estimates. Second, spatial downscaling or the integration of multi-source data (such as visible light or multispectral data) can help alleviate the impact of spatial heterogeneity on temperature estimates [46]. These approaches are expected to reduce the impact of spatiotemporal representativeness issues on the observation results and enhance the accuracy of thermal infrared remote sensing data.

## 5. Conclusions

This paper developed and applied a DJI UAV thermal image format conversion system, ThermoSwitcher, to address existing issues in thermal image stitching and processing and used it to analyze the spatiotemporal changes in the surface microthermal environment in two study areas in Nanjing, China. Firstly, the ThermoSwitcher system effectively resolved the problem of format conversion for DJI UAV thermal images, enabling the lossless conversion of JPG images while retaining temperature and geographic information and producing Geotiff format images that are easier to process and analyze. This tool not only improves the efficiency of thermal image processing but also enhances the potential application of UAV remote sensing technology in detailed studies of surface thermal environments. Secondly, the analysis results of the surface microthermal environment indicated significant heterogeneity in the spatial distribution of the surface temperatures within the study areas, with notable differences in temperatures across different local zones. High-temperature areas were mainly concentrated on sunlit artificial surfaces (such as asphalt pavements and building roofs), while low-temperature areas were primarily found in vegetation-covered and water body regions. A comparison of the temperature distribution before and after image stitching showed that, although the temperature distribution became more uniform after stitching, significant temperature differences between different zones remained, not affecting the analysis of temperature differences in local zones. This study not only provides technical support for neighbor-scale and detailed surface thermal environment monitoring, but also offers a data case for analyzing surface microthermal conditions. Future work should consider the zone scale effect and specific environmental characteristics, addressing the issues of temperature sampling accuracy and stitching quality of UAV thermal images. Furthermore, future work should also combine satellite, airborne, UAV remote sensing, and ground measurements to more fully understand and address the challenges posed by changes in the surface microthermal environment.

## Figures and Tables

**Figure 1 sensors-24-06267-f001:**
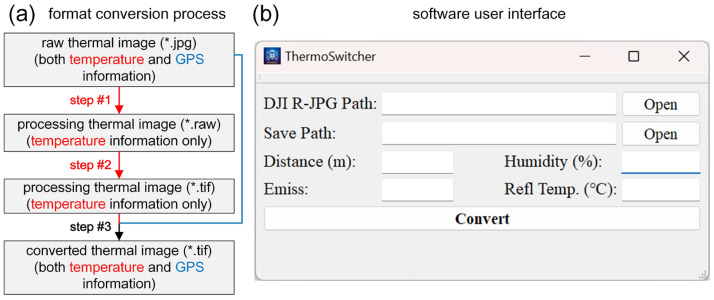
Format conversion process (**a**) and software user interface (**b**). In (**a**), the red and blue text and connection lines correspond to the temperature and GPS information components, respectively, and “*.jpg”, “*.raw” and “*.tif” stand for omitted file name and file format.

**Figure 2 sensors-24-06267-f002:**
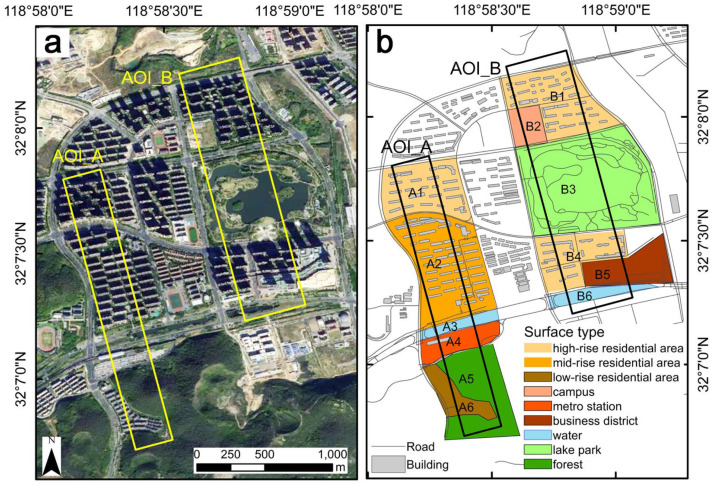
Study area. (**a**) Geographic location and satellite full-color images of the areas of interest A and B (AOI_A and AOI_B), (**b**) illustrations of the surface cover types of local zones in the areas of interest A and B.

**Figure 3 sensors-24-06267-f003:**
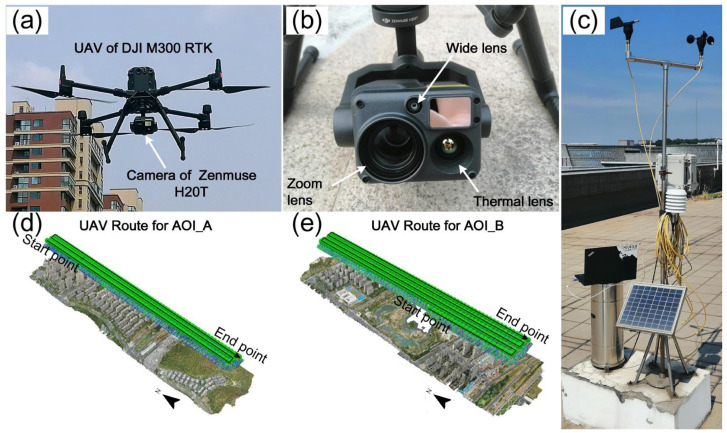
Observation equipment (**a**–**c**) and flight path (**d**,**e**).

**Figure 4 sensors-24-06267-f004:**
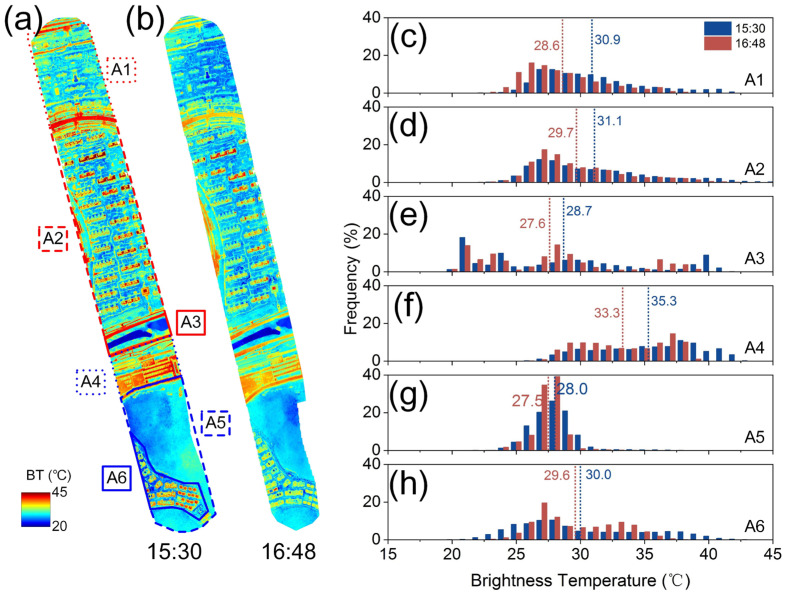
The spatial distribution (**a**,**b**) and frequency histograms (**c**–**h**) of the brightness temperatures in different zones at different observation times for AOI_A. (**a**,**b**) were sampled at 15:30 and 16:48 local time, respectively. The dashed lines and the values are the mean temperatures in (**c**–**h**).

**Figure 5 sensors-24-06267-f005:**
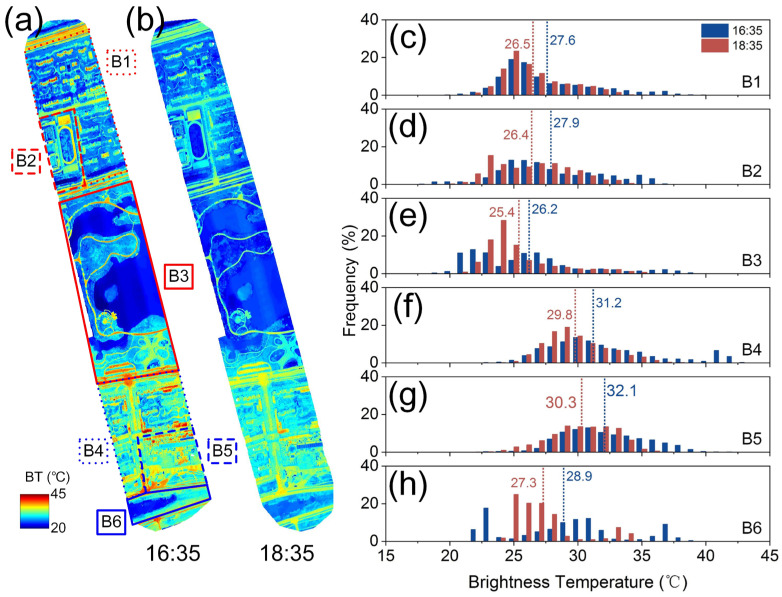
Similar to Figure 4, but for AOI_B. (**a**,**b**) were sampled at 16:35 and 18:35 local time, respectively. The dashed lines and the values are the mean temperatures in (**c**–**h**).

**Figure 6 sensors-24-06267-f006:**
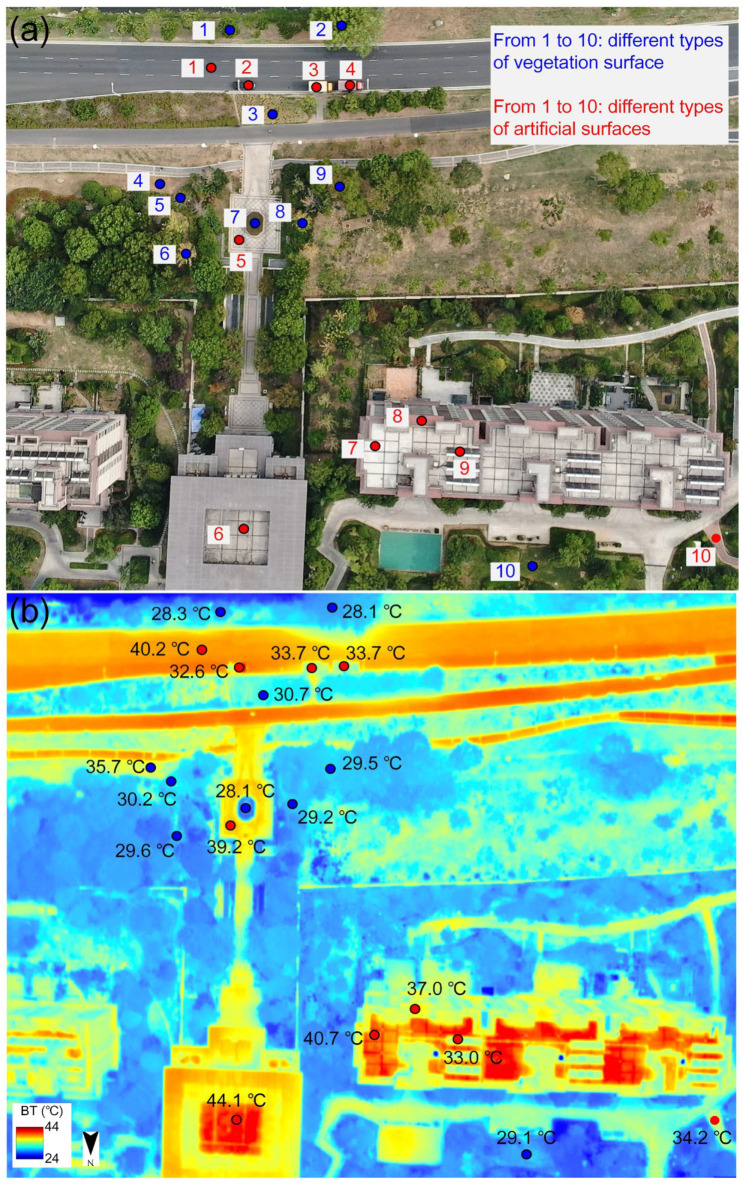
The RGB photo of different surface types (**a**) and the spatial distributions of the surface brightness temperatures at nadir at 15:33 (**b**). Specifically, the blue circles and numbers from 1 to 10 represent vegetated features, which are tree (willow), tree (maple), wilted tulip flowers, wilted grassland, tree (white elm), tree (ginkgo), shrub, tree (colored maple), tree (balsam fir), and grassland, respectively. The red circles and numbers from 1 to 10 represent artificial features, which are asphalt road, black metal car, yellow metal van, red metal van, stone pavement, glass roof, concrete roof, red-tile roof, solar water heater, and red-brick pavement, respectively.

**Figure 7 sensors-24-06267-f007:**
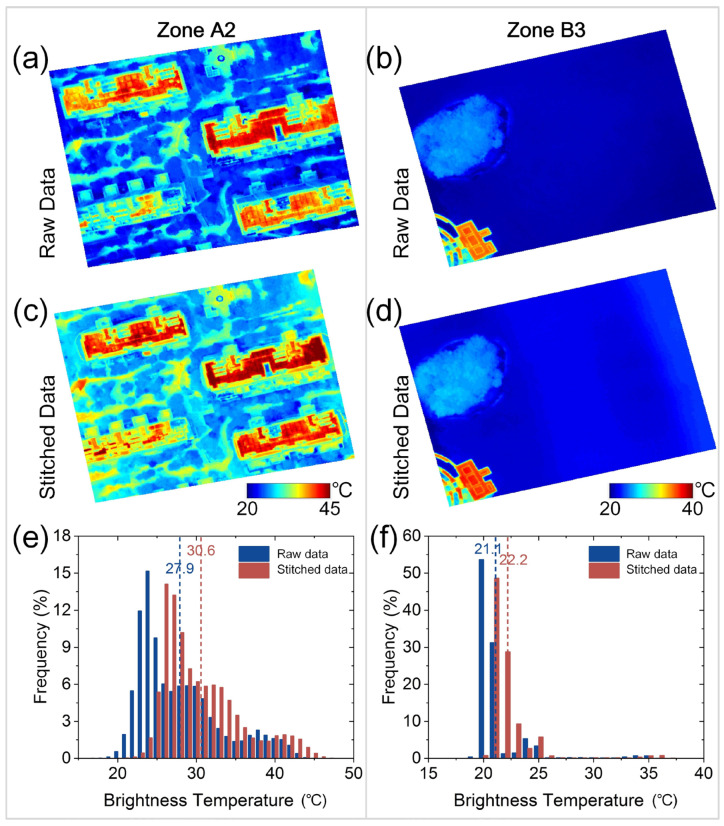
Comparison of thermal images and frequency histograms of brightness temperature distributions before (**a**,**b**) and after (**c**,**d**) stitching of building (A2) and lake (B3) zones. Dashed line and values are mean temperatures in (**e**,**f**).

**Table 1 sensors-24-06267-t001:** Meteorological parameters measured during the observation period.

Observation Area	Observation Time	Sun Position (Zenith Angle, Azimuth Angle)	Humidity (%)	Air Temperature (°C)
AOI_A	15:30	(50°, 259°)	27	35.7
AOI_B	16:35	(64°, 269°)	30	34.7
AOI_A	16:48	(66°, 271°)	31	34.3
AOI_B	18:35	(88°, 284°)	35	31.2

## Data Availability

The data used to support the findings of this study are available from the corresponding author upon request.

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
