# Peer review of "A UAV Thermal Imaging Format Conversion System and Its Application in Mosaic Surface Microthermal Environment Analysis"

_sensors, 2024, doi:10.3390/s24196267_

Round 1

Reviewer 1 Report

Comments and Suggestions for Authors

This article develops and applies the DJI drone thermal imaging format conversion system ThermoSwitch to solve existing problems in thermal imaging stitching and processing, and applies it to analyze the spatiotemporal changes of surface micro thermal environment in two research areas in Nanjing, China. From the author's description, the main contribution of this article is the proposal of ThermoSwitch, but this article does not provide a detailed introduction to ThermoSwitch, which raises many doubts. My comments and suggestions are as follows:

1.      Generally, abstracts should not include prospects for future work.

2.      At the end of the introduction, the contribution explanation is not clear. It is suggested to list it in points.

3.      It is recommended that the chapter structure of the paper be added to the last paragraph of the Introduction.

4.      In the paper, the section “Methods and Data” is not clear enough to describe the ThermoSwitch system, as if it were an introduction to the use of the ThermoSwitch system. The introduction of this section makes the paper look like a technical report.

5.      What are the specific contributions and innovations of this article? Is proposing a ThermoSwitch system the main contribution of this article? From the author's description, it can be seen that the system effectively solves the problem of thermal image conversion for DJI drones, achieving lossless conversion in JPG format while preserving temperature and geographic information, generating more easily processed and analyzed Geotif format images. If the ThermoWitcher system is the main contribution of this article, the author should consider opening it up and allowing readers to reproduce it, otherwise it is meaningless.

6.      From the introduction and analysis in this article, it can be seen that if the ThermoSwitch system can be made public, it would be of great help to the field of unmanned aerial vehicle remote sensing. Therefore, I look forward to the author's response to my above questions.

Author Response

Please see the 'Response to Reviewer 1' in the attachment

Reviewer 2 Report

Comments and Suggestions for Authors

This is a relevant study that supports the production of thermal data by UAV, ensuring uniformity among the acquired scenes. Despite its scientific importance, there was a gap in the methodology in detailing what the application actually does to stitch the images. Despite the specification of the parameters, what is the actual technique implemented in the application? Additionally, I understand that the validation method is not valid. It is necessary to have strategies to assess whether the temperature variability was corrected in line with the reality of the mapping. If validation was not conducted, I believe it should be a procedure for future research, as mentioned in the conclusion. I noticed that many discussions were superficial, considering the true objective of the research.

Specific issues in the text:

Abstract: I found the introduction disconnected from the article's objective. It did not answer whether the technique was able to stitch the scenes correctly.

Results indicated significant heterogeneity in the study area's temperature distribution, with high temperatures concentrated on sunlit artificial surfaces and low temperatures in building shadows, dense vegetation, and water areas. -> Too obvious, needs improvement. Were you able to map any urban pattern beyond vegetated and built areas?

Methods: Figure 1 – How did you obtain the calibration parameters? Did you have equipment on hand, or did the camera estimate them? How do these variables aid in the correction process? Mathematical formulas are important.

The figures are in poor resolution.

Figure 3 – What were the settings for image acquisition? Flight speed, 75% lateral and longitudinal overlap? Sun position at flight time, viewing angle. Why were Figures 3 and 4 organized with these 'Observation time' frames and not in sequence? What are these values highlighted in the graphs with dashed lines? Mean?"

Author Response

Please see the 'Response to Reviewer 2' in the attachment

Round 2

Reviewer 1 Report

Comments and Suggestions for Authors

Thank you very much for the author's response. The authors has carefully considered all the opinions and suggestions I have raised and provided a very detailed response to my concerns. Based on this, I suggest that the paper be considered for publication.

Before publishing the paper, the authors need to address the following two minor issues:

1.  There is an extra space after reference 27 on line 91.

2. Please divide the last paragraph of the introduction into several paragraphs to improve the readability of the article. The contribution of this article is set as a separate paragraph, and the structure of the article is set as a separate paragraph.

Author Response

Thanks very much for your critical comments and suggestions for improving the manuscript.

Comment 1:

  1. There is an extra space after reference 27 on line 91.

Response 1: Thank you for your thoughtful comments, we have deleted the extra space in the corresponding sentences. In addition, we have checked the full manuscript and all minor writing errors have been corrected.

Comment 2:

Please divide the last paragraph of the introduction into several paragraphs to improve the readability of the article. The contribution of this article is set as a separate paragraph, and the structure of the article is set as a separate paragraph.

Response 2: Thank you for your valuable advice. According to your comment, we have segmented the last paragraph of the introduction section. Please review the revised manuscript.

Reviewer 2 Report

Comments and Suggestions for Authors

Dear, the authors have made the suggested corrections. Best regards.

Author Response

Thanks very much for your critical comments and suggestions for improving the manuscript.